Morphological and molecular analyses of parasitic barnacles (Crustacea: Cirripedia: Rhizocephala) in Korea: preliminary data for the taxonomy and host ranges of Korean species

Jung Jibom 1
Yoshida Ryuta 2
Lee Damin 1
Park Joong-Ki jkpark@ewha.ac.kr 1
1 Division of EcoScience, Ewha Womans University , Seoul , South Korea
2 Tateyama Marine Laboratory, Marine and Coastal Research Center, Ochanomizu University , Tateyama , Chiba , Japan
Waiho Khor
Electronic publication date: 2021 Nov 12
Publication date: 2021
Volume: 9
Electronic Location ID: e12281
Received 2021 Jun 11; Accepted 2021 Sep 20
Copyright: ©2021 Jung et al.
Copyright year: 2021
Copyright holder: Jung et al.
License: This is an open access article distributed under the terms of the Creative Commons Attribution License, which permits unrestricted use, distribution, reproduction and adaptation in any medium and for any purpose provided that it is properly attributed. For attribution, the original author(s), title, publication source (PeerJ) and either DOI or URL of the article must be cited.
License URL: https://creativecommons.org/licenses/by/4.0/

Keywords: Morphology, Phylogenetic analysis, Taxonomy, Parasitic barnacles, Host range, Korean Rhizocephala

Funding: The National Institute of Biological Resources (NIBR) The Ministry of Environment (MOE) of the Republic of Korea NIBR202102203 NIBR202102108 Ewha Womans University RP-Grant 2021 The Hallyeohaesang National Park survey project supported by the Korea National Park Service This work was supported by a grant from the National Institute of Biological Resources (NIBR), funded by the Ministry of Environment (MOE) of the Republic of Korea (NIBR202102203; NIBR202102108), the RP-Grant 2021 of Ewha Womans University, and the Hallyeohaesang National Park survey project supported by the Korea National Park Service. This study was also supported by a Showa Seitoku Memorial Foundation. The funders had no role in study design, data collection and analysis, decision to publish, or preparation of the manuscript.

==============================
Morphological and molecular analyses of Korean rhizocephalan barnacle species were performed to examine their host ranges and taxonomy. Morphological examination and molecular analysis of mtDNA cox1, 16S, and nuclear 18S rRNA sequences revealed nine rhizocephalan species from three genera of the two families, Sacculinidae and Polyascidae. Phylogenetic analysis of molecular sequences revealed two new species candidates in the genus Parasacculina, and three Sacculina species (S. pilosella, S. pinnotherae, and S. imberbis) were transferred to the genus Parasacculina. Examination of host ranges revealed higher host specificity and lower infestation rates in Korean rhizocephalan species than rhizocephalans from other geographic regions. This is the first report of the taxonomy, species diversity, and host ranges of Korean parasitic rhizocephalan barnacles based on their morphological and molecular analyses. More information from extensive sampling of parasitic barnacles from a wide range of crustacean host species is necessary to fully understand their taxonomy, prevalence on decapod hosts, and phylogenetic relationships among major rhizocephalan taxa.

Introduction

The Rhizocephala comprises morphologically highly modified parasitic barnacles that use a wide range of crustaceans (mostly decapods) as their hosts, mostly parasites on decapods. Members of this group have complex life cycles, usually involving a series of pelagic larval stages followed by an endoparasitic interna stage and a reproductive externa stage where many organ systems (e.g., respiratory, digestive, sensory, and excretory systems) degenerate (Høeg, 1992; Øksnebjerg, 2000). In contrast to other crustacean species, they have a very simplified external structure (externa) and lack segmentation and appendages in the parasitic stage (Høeg & Lützen, 1995). Due to the simplified morphology of the externa, previous taxonomic studies of rhizocephalans have been based largely on larval morphology and the fine structure of externa observed from paraffin sectioning (Yoshida et al., 2011; Kobayashi et al., 2018), with further validation through DNA barcoding analysis (Yoshida et al., 2012; Yoshida, Hirose & Hirose, 2014; Høeg et al., 2019; Jung, Yoshida & Kim, 2019). Very recently, Høeg et al. (2019) modified the taxonomic system of Rhizocephala based on molecular phylogenetic analysis of 18S rDNA sequences, and additional molecular-based taxonomy of rhizocephalan barnacles was updated in Chan et al. (2021).

Since the first report of rhizocephalan barnacles (Krüger, 1912; Høeg & Lützen, 1985), northwestern Pacific species have been reported from southeast Russia (Korn et al., 2020), China (Li et al., 2015), Taiwan (Tu, Chan & Jeng, 2009; Yoshida et al., 2012), Japan (Shiino, 1943; Utinomi & Kikuchi, 1966; Nagasawa, Lützen & Kado, 1996), and Korea (Jung, Yoshida & Kim, 2019). Although Jung, Yoshida & Kim (2019) described 10 species of peltogastrid barnacles from 17 hermit crab species in Korea, the species diversity, distribution, taxonomy, and host range of other rhizocephalan parasitic barnacles are still largely unknown in Korea. In this study, we characterized nine rhizocephalan species, including two new cryptic species candidates, based on morphological examination and molecular analyses of mitochondrial (cytochrome c oxidase subunit I and 16S) and nuclear 18S rDNA sequences. In addition to the taxonomic accounts of the Korean species, we also investigated the prevalence of parasitic barnacles in decapod hosts and phylogenetic relationships among rhizocephalan species.

Methods

We examined the abdomens of 3,262 individuals of 25 Korean decapod host species collected from 16 sampling sites in Korea. In addition, 12 Japanese rhizocephalans from 11 sampling sites were obtained for comparative molecular study (Tables 1, 2). Korean voucher specimens in this study were deposited in the National Institute of Biological Resources (NIBR) and Honam the National Institute of Biological Resources (HNIBR). Japanese voucher specimens in this study were deposited in the Ryukyu University Museum, Fujukan, University of the Ryukyus, Okinawa, Japan (RUMF), and Coastal Beach of Natural History Museum and Institute, Chiba, Japan (CMNH).

Table 1 Individual number and infestation rate (%) of Korean decapod species by rhizocephalan parasitic barnacles examined in this study.

Host decapod species	Total number of individuals examined	Number of individuals infested	Infestation rate	
Alpheus bisincisus	16	0	0.0%	
Arcotheres sinensis	4	2	50.0%	
Eualus sinensis	16	0	0.0%	
Gaetice depressus	1400	14	1.0%	
Helicana japonica	28	0	0.0%	
Hemigrapsus penicillatus	50	0	0.0%	
Hemigrapsus sanguineus	114	9	7.9%	
Hemigrapsus takanoi	145	1	0.7%	
Ilyoplax dentimerosa	10	0	0.0%	
Ilyoplax pusilla	57	0	0.0%	
Laomedia astacina	15	0	0.0%	
Macromedaeus distinguendus	74	8	10.8%	
Macrophthalmus (Mareotis) japonicus	45	0	0.0%	
Neotrypaea japonica	35	0	0.0%	
Pachygrapsus crassipes	1	1	100.0%	
Pagurus lanuginosus	65	0	0.0%	
Pagurus minutus	811	0	0.0%	
Pagurus nigrofascia	28	0	0.0%	
Palaemon serrifer	17	0	0.0%	
Parasesarma pictum	61	0	0.0%	
Pugettia intermedia	14	1	7.1%	
Scopimera globosa	25	0	0.0%	
Sestrostoma balssi	16	0	0.0%	
Stenalpheops anacanthus	61	0	0.0%	
Upogebia major	154	2	1.3%	
Total	3262	38	1.2%	

All rhizocephalan specimens were fixed in 95% ethanol and subjected to morphological examination and molecular analysis. For morphological analysis, the externa and mantle were examined using an MZ8 dissection microscope (Leica, Wetzlar, Germany). Photographs were taken with a D200 digital camera (Nikon, Tokyo, Japan). Carapace length (cl) of the host decapod was measured as the length from the tip of the rostrum to the midpoint of the posterior margin of the carapace using a CD6CSX digital caliper (Mitutoyo, Kawasaki, Japan) to the nearest 0.1 mm.

For molecular analysis, the lateral end of the externa tissue of each rhizocephalan specimen was excised for total genomic DNA extraction using the QIAamp DNA Micro Kit (QIAGEN, Hilden, Germany). Universal primers LCO1490 (5′-GGTCAACAAATCATAAAGATATTGG-3′) and HCO2198 (5′-TAAACTTCAGGGTGACCAAAAAATCA-3′) were used to amplify a fragment of mitochondrial cytochrome c oxidase subunit I (cox1) (Folmer et al., 1994). To amplify the mitochondrial 16S rDNA gene, 16SH2 (5′-AGATAGAAACCAACCTGG-3′) and 16SL2 (5′-TGCCTGTTTATCAAAAACAT-3′) primers (Schubart, Neigel & Felder, 2000) were used. For PCR amplification of 18S rDNA, 18S-329R (TAATGATCCTTCCGCAGGTT) and 18S-AF (CAGCMGCCGCGGTAATWC) primers were used (Spears, Abele & Kim, 1992). Polymerase chain reaction (PCR) was performed in reaction volumes of 50 µL that included 2 µL DNA template, 5 µL 10 x Ex Taq Buffer, 2 µL of each primer (10 µM), 0.25 µL Go Taq DNA polymerase (Promega, Madison City, WI, USA), 2.5 µL dNTP mix (10 mM), and 35.75 µL distilled H2O. PCR amplification was performed using the following steps: 5 min denaturation at 94 °C followed by 35 cycles of 30 s at 94 °C, 1 min at 52 °C, 1 min at 72 °C, and a final extension of 7 min at 72 °C. PCR products were visualized on 1% agarose gels and sequenced with an ABI PRISM 3730xl DNA analyzer (Applied Biosystems, Foster City, CA, USA). Nucleotide sequences of the three gene fragments (mtDNA cox1, 16S, and nuclear 18S rDNA) were analyzed and edited using Geneious v. 9.1.8 (Kearse et al., 2012) and aligned using ClustalW in the MEGA10 program (Kumar et al., 2018). Nucleotide sequences were deposited in GenBank (mtDNA cox1: MZ216468–MZ216513; 16S: MZ215675–MZ215720; 18S rDNA: MZ215557–MZ215602). Forty-six additional rhizocephalan sequences of Sacculinidae and Polyascidae species available in GenBank were downloaded and included in the phylogenetic analyses (Table 2).

Table 2 GenBank accession numbers, geographic information, and host species of rhizocephalan species used for phylogenetic analysis in this study.

Species	Host species	Location	Specimen number	cox1 GenBank accession no	16S rDNA GenBank accession no	18S rDNA GenBank accession no	
Boschmaella japonica*	Chthamalus challengeri	Jôgashima, Japan	ZMUC CRU-3877			AY265369 	
Briarosaccus regalis*	Paralithodes camtschaticus	Alaska, USA		KR812178	KR812157		
Clistosaccus paguri*		Germany		KT208500			
Heterosaccus californicus*	Loxorhynchus grandis	CA, USA			AY520756		
Santa Barbara, USA 	ZMUC CRU-3875			AY265359 	
Heterosaccus dollfusi*	Charybdis longicollis	Mediterranean			FJ481949		
Portunus pelagicus	Sri Lanka		KY030832			
Heterosaccus lunatus*	Charybdis callianassa	Moreton Bay, Australia		DQ059778	FJ481947		
Heterosaccus papillosus*	Charybdis bimaculata	Korea			FJ481948		
Loxothylacus panopaei*	Rhithropanopeus harrisii	Neuse River, USA	ZMUC CRU-3876		FJ481956	AY265364 	
xanthoid crabs	USA		HQ848065			
Loxothylacus texanus*	xanthoid crabs	USA		HQ848066			
Mycetomorpha vancouverensis*		Alaska, USA			MH974513		
Parasacculina beauforti*	Scylla olivacea	Malaysia		KX426583			
Parasacculina compressa*	Ozius tuberculosus	Panglao, Philippines			KF561276		
Parasacculina granifera*	Portunus pelagicus	Moreton Bay, Australia		DQ059779			
Parasacculina imberbis comb. nov.	Pachygrapsus crassipes	Namhae, Korea	Korea 4	MZ216470	MZ215675	MZ215557	
Shirosaki, Japan*	S2	AB197804			
Parasacculina leptodiae*	Leptodius affinis	Labrador, Singapore	ZMUC CRU-3870		FJ481952	AY265365 	
Parasacculina oblonga*	Cyclograpsus intermedius	Amakusa, Japan	G4028	DQ059780			
Tomioka, Japan	ZMUC CRU-3871		FJ481953	AY265367 	
Parasacculina pilosella comb. nov.	Pugettia intermedia	Sacheon, Korea	VSJAIV0000000010 (Korea 3)		MZ215679	MZ215561	
Parasacculina pinnotherae comb. nov.	Arcotheres sinensis	Imabari, Japan	CMNH-ZC-02762 (Japan 9)	MZ216499	MZ215676	MZ215558	
Busan, Korea	Korea 1	MZ216468	MZ215677	MZ215559	
Korea 2	MZ216469	MZ215678	MZ215560	
Parasacculina sinensis*	Leptodius affinis	Hong Kong, China	ZMUC CRU-3874			AY265360 	
Parasacculina shiinoi	Upogebia major	Namhae, Korea	Korea 25	MZ216486		MZ215562	
Korea 26	MZ216487	MZ215680	MZ215563	
Japan*			KF539761		
Parasacculina sp. 1	Macromedaeus distinguendus	Sacheon, Korea	Korea 15	MZ216480	MZ215683	MZ215566	
Korea 16	MZ216481	MZ215684	MZ215567	
Korea 17	MZ216482	MZ215685	MZ215568	
Korea 20	MZ216511	MZ215686	MZ215569	
Korea 21	MZ216484	MZ215687	MZ215570	
Yeosu, Korea	Korea 14	MZ216479	MZ215682	MZ215565	
Korea 19				
Parasacculina sp. 2	Macromedaeus distinguendus	Sacheon, Korea	Korea 18	MZ216483	MZ215688	MZ215571	
Parasacculina sp.	Guinusia dentipes	Tateyama, Japan	CMNH-ZC-02756 (Japan 6)	MZ216497	MZ215681	MZ215564	
Parasacculina yatsui	Hemigrapsus sanguineus	Taean, Korea	VSJAIV0000000011 (Korea 36)	MZ216507	MZ215694	MZ215576	
Toga Bay, Japan*			MG822656		
Tongyeong, Korea	Korea 38		MZ215695	MZ215577	
Korea 39	MZ216509	MZ215696	MZ215578	
Pachygrapsus crassipes	Misaki, Japan	CMNH-ZC-02770 (Japan13)	MZ216503	MZ215691	MZ215574	
Tateyama, Japan	CMNH-ZC-02764 (Japan10)	MZ216500	MZ215689	MZ215572	
Katsuura, Japan	CMNH-ZC-02758 (Japan 7)		MZ215692		
Tateyama, Japan	CMNH-ZC-02760 (Japan 8)	MZ216498	MZ215693	MZ215575	
Shirosaki, Japan*	S13	AB197810			
Tateyama, Japan	CMNH-ZC-02768 (Japan 12)	MZ216502	MZ215690	MZ215573	
Peltogasterella gracilis*	Pagurus filholi	Gyeongju, Korea	MADBK 160707_039	MK604152			
Pagurus pectinatus	Busan, Korea	MADBK 430103_002		MK604172		
Peltogaster postica*	Pagurus angustus	Chisi, Taiwan	NMNS-6795-003		AB778096		
Pagurus filholi	Jeju, Korea	MADBK 430102_002	MK604144			
Peltogaster reticulata*	Pagurus minutus	Namhae, Korea	MADBK 160706_065		MK604167		
Vostok Bay, Russia		MN193579			
Polyascus gregarius*	Eriocheir japonica 	Maruyama, Japan	ZMUC CRU-3869			AY265363 	
Polyascus cf. gregarius	Hemigrapsus sanguineus	Namhae, Korea	VSJAIV0000000013 (Korea 27)	MZ216488	MZ215697	MZ215579	
Korea 30	MZ216513	MZ215700	MZ215582	
Sacheon, Korea	Korea 29	MZ216490	MZ215699	MZ215581	
Taean, Korea	VSJAIV0000000012 (Korea 37)	MZ216508	MZ215703		
Yeosu, Korea	VSJAIV0000000014 (Korea 28)	MZ216489	MZ215698	MZ215580	
Korea 31	MZ216491	MZ215701	MZ215583	
Hemigrapsus takanoi	Namhae, Korea	Korea 32	MZ216492	MZ215702	MZ215584	
Polyascus planus	Grapsus albolineatus	Nakagusuku, Okinawa, Japan	RUMF-ZC-7303 (Japan 2)	MZ216494	MZ215705	MZ215586	
Kenting, Taiwan*	ZMUC CRU-3872		FJ481954	AY265368 	
Metopograpsus messor	Nago, Okinawa, Japan	RUMF-ZC-7305 (Japan 3)	MZ216495	MZ215706	MZ215587	
Iriomote, Japan	RUMF-ZC-7309 (Japan 5)	MZ216496	MZ215707	MZ215588	
Nago, Okinawa, Japan	RUMF-ZC-7301 (Japan 1)	MZ216493	MZ215704	MZ215585	
Polyascus polygeneus*	Hemigrapsus sanguineus	Ôyano, Japan	ZMUC CRU-3873			AY265362 	
Sacculina carcini*	Carcinus maenas	Gullmar Fjord, Sweden	ZMUC CRU-3867		FJ481957	AY265366 	
Sacculina confragosa	Gaetice depressus	Namhae, Korea	Korea 11	MZ216477	MZ215709	MZ215590	
VSJAIV0000000015 (Korea 12)	MZ216478	MZ215710	MZ215591	
Sacheon, Korea	Korea 5	MZ216471	MZ215715	MZ215597	
Korea 6	MZ216472	MZ215716	MZ215598	
Korea 7	MZ216473	MZ215717	MZ215599	
Korea 8	MZ216474	MZ215718	MZ215600	
Korea 9	MZ216475	MZ215719	MZ215601	
Korea 10	MZ216476	MZ215708	MZ215589	
Korea 23	MZ216485	MZ215712	MZ215593	
Korea 24	MZ216512		MZ215594	
Tongyeong, Korea	VSJAIV0000000016 (Korea 33)	MZ216504	MZ215713	MZ215595	
Korea 34	MZ216505	MZ215714	MZ215596	
Korea 35	MZ216506			
Yeosu, Korea	Korea 13	MZ216510	MZ215711	MZ215592	
Pachygrapsus crassipes*	Shirama, Japan	ZMUC CRU-3868			AY265361 	
Shirosaki, Japan	S22	AB197803			
Sacculina insueta*	Ptychognathus riedelii	Kawasan, Philippines			KF561274		
Sacculina upogebiae*					KF539762		
Sacculinidae sp.	Thalamita sp.	Tateyama, Japan	CMNH-ZC-02766 (Japan 11)	MZ216501	MZ215720	MZ215602	
Sesarmaxenos gedehensis*	Sesarmops sp.	Kawasan, Philippines			KF561270		
Sylon hippolytes*				MG313989			
Notes.

* sequences derived from GenBank.

Phylogenetic relationships among rhizocephalan species were inferred for each of the three genes using maximum likelihood (ML) analysis and Bayesian inference (BI) implemented in RaxML version 8 (Stamatakis, 2014) and MrBayes v3.2.6 (Ronquist, Huelsenbeck & Teslenko, 2011), respectively. Phylogenetic trees were modified by MEGA 10. Maximum likelihood analyses of cox1, 16S, and 18S rDNA sequences were performed based on the Tamura-Nei (TN93) (Tamura & Nei, 1993), general time reversible (Tavaré, 1986), and Kimura 2-parameter (Kimura, 1980) models, respectively, with a gamma distribution (+G) and invariable sites (+I) rate categories based on Bayesian Information Criterion (BIC) scores model using the Model Selection option of MEGA10. The robustness of individual nodes in the ML trees was assessed by analysis of 1,000 bootstrap replications. Interspecific and intraspecific sequence divergences were estimated based on the K2P distance matrix in MEGA10.

Results

Based on morphological examination (shape and number of externae and mantle aperture) and mitochondrial sequence information, we identified 38 rhizocephalan individuals belonging to nine species, three genera, and two families isolated from eight decapod hosts species collected from 16 sites (Fig. 1; Table 2). All rhizocephalans identified by this study except Parasacculina pinnotherae comb. nov. were first reported from Korea. Detailed information regarding the Korean rhizocephalan species and their externa morphology is provided in Table 3.

Taxonomic accounts and morphological features of Korean rhizocephalan species

Sacculinidae Lilljeborg, 1861	
Sacculina Thompson, 1836	

Sacculina confragosa Boschma1933 (Fig. 2A)

Materials examined: on Gaetice depressus: 1 ind., Sacheon (34.9 N 128.1 E), Korea 5, host: ♀, cl 11.5 mm; 1 ind. (2 externa), Sacheon (34.9 N 128.1 E), Korea 6, host: ♀, cl 11.8 mm; 1 ind., Sacheon (34.9 N 128.0 E), Korea 7, host: ♀, cl 13.7 mm; 1 ind., Sacheon (34.9 N 128.0 E), Korea 8, host: ♀, cl 9.1 mm; 1 ind., Sacheon (34.9 N 128.1 E), Korea 9, host: ♀, cl 10.3 mm; 1 ind., Sacheon (34.9 N 128.1 E), Korea 10, host: ♀, cl 11.8 mm; 1 ind., Namhae (34.7 N 127.9 E), Korea 11, host: ♂, cl 11.9 mm, feminization; 1 ind. (2 externa), Namhae (34.7 N 127.9 E), VSJAIV0000000015, Korea 12, host: ♀, cl 7.7 mm; 1 ind., Yeosu (34.7 N 127.8 E), Korea 13, host: ♂, cl 11.2 mm, feminization; 1 ind., Sacheon (34.9 N 128.0 E), Korea 23, host: ♂, cl 12.0 mm; 1 ind., Sacheon (34.9 N 128.0 E), Korea 24, host: ♂, cl 13.9 mm; 1 ind., Tongyeong (34.8 N 128.4 E), VSJAIV0000000016, Korea 33, host: ♀, cl 7.1 mm; 1 ind., Tongyeong (34.8 N 128.4 E), Korea 34, host: ♂, cl 14.9 mm; 1 ind., Tongyeong (34.6 N 128.5 E), Korea 35, host: ♂, cl 13.5 mm.

Figure 1 Map showing the collection sites of the Korean rhizocephalan species.

Numbered circles indicate sampling localities where rhizocephalan species were sampled in this study (blue) and Jung, Yoshida & Kim (2019); black). 1, Sacculina confragosa; 2, Parasacculina imberbis; 3, Parasacculina pilosella; 4, Parasacculina pinnotherae; 5, Parasacculina shiinoi; 6, Parasacculina sp. 1; 7, Parasacculina sp. 2; 8, Parasacculina yatsui; 9, Polyascus cf. gregarius; 10, Peltogaster lineata; 11, Peltogaster postica; 12, Peltogaster aff. ovalis; 13, Peltogaster aff. reticulatus; 14, Peltogaster sp. 1; 15, Peltogaster sp. 2; 16, Peltogaster sp. 3; 17, Peltogasterella gracilis.

Table 3 Morphological features of the externa of nine Korean rhizocephalans.

Species	Externa	Mantle aperture	
	Shape	Externa number	Projection	Opening	
Sacculina confragosa	Wrinkled flat cordiform	Single or double	Elevated tube	Circular	
Parasacculina imberbis comb. nov.	Smooth round-rectangular	Single			
Parasacculina pilosella comb. nov.	Smooth and slightly flat oval	Single	Flat	Circular	
Parasacculina pinnotherae comb. nov.	Smooth or slightly wrinkled flat oval or cordiform	Single or double	Elevated	Dot shaped	
Parasacculina shiinoi	Smooth oval	Single			
Parasacculina sp. 1	Smooth or slightly wrinkled oval	Single	Elevated	Circular	
Parasacculina sp. 2	Smooth oval	Single	Slightly elevated	Circular	
Parasacculina yatsui	Smooth or slightly wrinkled flat oval or flat cordiform	Single	Elevated	Slit shaped	
Polyascus cf. gregarius	Smooth or slightly wrinkled flat cordiform	Single	Elevated	Slit shaped	

Figure 2 Externae of Korean rhizocephalans.

Red arrow: mantle, scale bar: two mm. (A) Sacculina confragosa. (B) Parasacculina pilosella comb. nov. (C) Parasacculina pinnotherae comb. nov. (D) Parasacculina sp. 1. (E) Parasacculina sp. 2. (F) Parasacculina yatsui. (G) Polyascus cf. gregarius. Externae of some specimens (B, D–G) were incomplete, in case they were used for molecular analysis (B, D–G).

Host species: G. depressus, Pachygrapsus crassipes (Grapsidae), Cyclograpsus intermedius (Varunidae).

Distribution: Japan, Korea.

Diagnosis of the externa: whole externa mostly single and occasionally double, wrinkled cordiform with flat half-oval-shaped left and right lobes divided by an outer mid-groove and inner mid-ridge; outermost part of the robe wrinkled. Mantle well elevated, tube-shaped, and vertically slightly wrinkled with a circular opening at the extremity.

Remarks: Morphological characteristics of the examined materials correspond with their original description (Boschma, 1933) except for the number of externa. Some of our specimens (Korea 6, Korea 12) had double externae (15% of total examined individuals), whereas others had a single externa. This type of variation in the number of externa has been reported in a previous study (Shiino, 1943). This species is found most abundantly parasitizing medium-sized individuals of host crab species. Further study is needed to determine if this species is a predominant parasitic form on medium-sized host individuals.

Parasacculina imberbis (Shiino, 1943) comb. nov.

Polyascidae Høeg & Glenner in Høeg et al., 2019

Parasacculina Høeg & Glenner in Høeg et al., 2019

Materials examined: 1 ind., Namhae (34.7 N 128.0 E), Korea 4, host: ♀, cl 12.5 mm.

Host species: Pachygrapsus crassipes (Grapsidae).

Distribution: Japan, Korea.

Diagnosis of the externa: whole externa smooth, single, with a rounded-rectangular shape.

Remarks: The examined specimen had a single externa, but we were not able to examine morphological characteristics in more detail due to the immature stage of the specimen. Species identification of this specimen and its taxonomic placement in the genus Parasacculina were based on molecular analyses of mtDNA cox 1, 16S, and nuclear 18S gene sequences (Figs. 3A–3C; see Discussion for more details).

Figure 3 Phylogenetic tree of cox1 from rDNA from Korean rhizocephalan species using Maximum likelihood and Bayesian inference methods.

Values on nodes indicate maximum likelihood bootstrap support/Bayesian posterior probability. Sequences from Korean species determined in this study are indicated in blue. *: sequences derived from GenBank, **: Japanese sequences obtained in this study.

Figure 4 Phylogenetic tree of 16S rDNA from Korean rhizocephalan species using Maximum likelihood and Bayesian inference methods.

Values on nodes indicate maximum likelihood bootstrap support/Bayesian posterior probability. Sequences from Korean species determined in this study are indicated in blue. *: sequences derived from GenBank, **: Japanese sequences obtained in this study.

Figure 5 Phylogenetic tree of 18S rDNA from Korean rhizocephalan species using Maximum likelihood and Bayesian inference methods.

Values on nodes indicate maximum likelihood bootstrap support/Bayesian posterior probability. Sequences from Korean species determined in this study are indicated in blue. *: sequences derived from GenBank, **: Japanese sequences obtained in this study.

Figure 6 A vertical bar chart showing the individual number of nine Korean rhizocephalan species found from their hosts.

Parasacculina pilosella (Kampen & Boschma, 1925) comb. nov. (Fig. 2B)

Material examined: 1 ind., Sacheon (34.9 N 128.1 E), VSJAIV0000000010, Korea 3, host: ♂, cl 13.7 mm.

Host species: Pugettia intermedia (Epialtidae).

Distribution: Indonesia, Japan, Korea.

Diagnosis of the externa: whole externa smooth, single, and slightly flat and oval. Mantle flat and vertically slightly wrinkled with a circular opening at the extremity.

Remarks: Four Sacculina species (S. muricata Boschma, 1931, S. pugettiae (Shiino, 1943), S. reinhardi (Boschma, 1955), and S. pilosella) were previously reported to parasitize Pugettia spp.. Morphological characteristics of the examined specimen correspond with the original description of S. pillosella (Van Kampen & Boschma, 1925). However, phylogenetic analysis of mtDNA 16S and 18S rDNA sequences showed that this species is nested within Parasacculina species (Figs. 3B and 3C), separated from Sacculina species (S. confragosa, S. upogebiae, and S. carcini). Therefore, we consider this species a member of the genus Parasacculina (see Discussion for more details).

Parasacculina pinnotherae (Shiino, 1943) comb. nov. (Fig. 2C)

Materials examined: 1 ind., Busan (35.2 N 129.2 E), Korea 1, host: ♀, cl 9.1 mm, in the mussel; 1 ind. (2 externa), Busan (35.2 N 129.2 E), Korea 2, host: ♂, cl 6.9 mm, in the mussel.

Host species: Arcotheres sinensis (Pinnotheridae).

Distribution: Japan, Korea.

Diagnosis of the externa: whole externa smooth or slightly wrinkled, single or double, and flat oval or cordiform in shape; each outer-posterior margin elevated into a conical shape. Mantle slightly elevated and vertically wrinkled with a small round opening at the extremity.

Remarks: Two Sacculina species (S. pertenuis (Boschma, 1933) and S. pinnotherae) have been reported to be parasitic on Pinnotheres spp.. Morphological characters of examined specimens correspond with the original description of S. pinnotherae (Shiino, 1943).

However, phylogenetic analysis of mtDNA cox 1, 16S, and nuclear 18S rDNA sequences placed this species within the genus Parasacculina (Figs. 3A–3C), not in the genus Sacculina. Therefore, we treated this species as a member of the genus Parasacculina (see Discussion for more details). The host crab (Arcotheres sinensis) is known to parasitize bivalves, so P. pinnotherae comb. nov. is a secondary parasite that is rare in the ocean (McDermott, 2009).

Parasacculina shiinoi (Lützen et al., 2016)

Materials examined: 1 ind., Namhae (34.9 N 127.9 E), Korea 25, host: cl 8.8 mm; 1 ind., Namhae (34.9 N 127.8 E), Korea 26, host: cl 11.1 mm.

Host species: Upogebia major (Upogebiidae).

Distribution: Japan, Korea.

Diagnosis of the externa: whole externa smooth, single, oval in shape.

Remarks: The examined specimens had a single externa, but detailed morphological characteristics could not be determined because of the immaturity of the specimens examined. Lützen et al. (2016) reported that Sacculina upogebiae parasitizes Upogebia species. Molecular analysis of mtDNA 16S rDNA sequences revealed that this species grouped with P. shiinoi (GenBank accession no: KF539761: Fig. 3B) with very high sequence identity (98.9%).

Parasacculina sp. 1 (Fig. 2D)

Materials examined: 1 ind., Yeosu (34.7 N 127.8 E), Korea 14, host: ♀, cl 7.6 mm; 1 ind., Sacheon (34.9 N 128.0 E), Korea 15, host: ♀, cl 8.0 mm; 1 ind., Sacheon (34.9 N 128.0 E), Korea 16, host: ♂, cl 9.7 mm; 1 ind., Sacheon (34.9 N 128.0 E), Korea 17, host: ♂, cl 9.9 mm; 1 ind., Yeosu (34.7 N 127.8 E), Korea 19, host: ♂, cl 14.6 mm; 1 ind., Sacheon (34.9 N 128.0 E), Korea 20, host: ♂, cl 14.1 mm; 1 ind., Sacheon (34.9 N 128.0 E), Korea 21, host: ♂, cl 17.7 mm.

Host species: Macromedaeus distinguendus (Xanthidae).

Distribution: Korea.

Diagnosis of the externa: whole externa single, smooth or slightly wrinkled, and oval in shape. Mantle large, elevated, and vertically wrinkled with circular opening at the extremity.

Remarks: Parasacculina leptodiae and P. sinensis have been reported to be parasites of Leptodius affinis, the most phylogenetically similar host species to M. distinguendus among the currently known hosts of Rhizocephala. However, the specimens examined in this study differ in morphology and molecular sequences from P. leptodiae and P. sinensis. This species has a single externa, whereas P. leptodiae has multiple externae. In addition, this species has a large, elevated mantle aperture, but P. leptodiae and P. sinensis have a flat mantle (Guérin-Ganivet, 1911; Boschma, 1933). Phylogenetic analysis clearly showed that the cox 1, 16S, and 18S rDNA sequences of this species are different from those of P. leptodiae and P. sinensis and all other Parasacculina species included in the analyses (Figs. 3A–3C). Therefore, we considered this species to be a new species candidate of the genus Parasacculina (see Discussion for more details).

Parasacculina sp. 2 (Fig. 2E)

Material examined: 1 ind., Sacheon (34.9 N 128.1 E), Korea 18, host: ♂, cl 13.5 mm.

Host species: Macromedaeus distinguendus (Xanthidae).

Distribution: Korea.

Diagnosis of the externa: whole externa smooth, single, and oval in shape. Mantle slightly elevated with circular opening at extremity.

Remarks: Previously, P. leptodiae and P. sinensis were known to be parasites of L. affinis, the most phylogeneically similar host species to M. distinguendus among the currently known hosts of Rhizocephala. However, the examined specimen of this species differed in morphology and molecular sequence from all Parasacculina species including Parasacculina sp. 1. This species has one externa compared to the multiple externae of P. leptodiae. In addition, this species has a slightly elevated mantle aperture, whereas P. leptodiae and P. sinensis have a flat mantle (Guérin-Ganivet, 1911; Boschma, 1933), and Parasacculina sp. 1 has a well-elevated mantle. Furthermore, phylogenetic analysis of mtDNA cox 1, 16S, and nuclear 18S rDNA sequences distinguished this species from other Parasacculina species with 18.1–30.6% sequence divergence in cox1, 14.0–27.5% sequence divergence in 16S rDNA, and 1.9−4.2% sequence divergence in 18S rDNA (Figs. 3A–3C). Therefore, we treated this species as a new species candidate of the genus Parasacculina (see Discussion for more details).

Parasacculina yatsui (Boschma, 1936) (Fig. 2F)

Materials examined: on Hemigrapsus sanguineus: 1 ind., Yeosu (34.1 N 127.3 E), VSJAIV0000000011, Korea 36, host: ♀, cl 22.3 mm; 1 ind., Tongyeong (34.8 N 128.4 E), Korea 38, host: ♂, cl 16.9 mm, feminization; 1 ind., Tongyeong (34.6 N 128.5 E), Korea 39, host: ♂, cl 14.2 mm, feminization.

Host species: Pachygrapsus crassipes (Grapsidae), H. sanguineus (Varunidae).

Distribution: Japan, Korea.

Diagnosis of the externa: whole externa smooth or slightly wrinkled, single, and cordiform with flat half-oval-shaped left and right lobes divided by outer mid-groove and inner mid-ridge; outermost part of the robe wrinkled. Mantle tube-shaped, elevated, and slightly wrinkled with slit opening at extremity.

Polyascus cf. gregarius (Okada & Miyashita, 1935) (Fig. 2G)

Polyascus Glenner, Lützen & Takahashi, 2003

Materials examined: on H. sanguineus: 1 ind., Namhae (34.7 N 127.9 E), VSJAIV0000000013, Korea 27, host: ♀, cl 18.4 mm; 1 ind., Yeosu (34.7 N 127.8 E), VSJAIV0000000014, Korea 28, host: ♂, cl 29.8 mm, feminization; 1 ind., Sacheon (34.9 N 128.1 E), Korea 29, host: ♀, cl 9.8 mm; 1 ind., Namhae (34.9 N 127.8 E), Korea 30, host: ♂, cl 16.7 mm, feminization; 1 ind., Yeosu (34.7 N 127.8 E), Korea 31, host: ♂, cl 8.9 mm, feminization; 1 ind., Taean (36.8 N 126.1 E), Korea 37, host: ♂, cl 14.1 mm, feminization.

on H. takanoi: 1 ind., Namhae (34.9 N 127.9 E), Korea 32, host: ♂, cl 9.5 mm.

Host species: Hemigrapsus sanguineus, H. takanoi, Eriocheir japonica (Varunidae).

Distribution: Japan, Korea.

Diagnosis of the externa: whole externa smooth or slightly wrinkled, single, and flat-cordiform shaped with flat half-oval-shaped left and right lobes divided by an outer mid-groove and inner mid-ridge; outermost part of the robe smooth or slightly wrinkled. Mantle tube-shaped, elevated, and vertically wrinkled with slit-shaped opening at the extremity.

Remarks: Morphological characteristics of the examined materials correspond with their original description (Okada & Miyashita, 1935) except for the number of externa and the host species. All specimens examined in this study had a single externa, whereas P. gregarius has multiple externae. In addition, the host species (H. sanguineus and H. takanoi) differ from the host species reported for P. gregarius, namely E. sinensis. Nevertheless, this species is likely P. gregarius because 18S rDNA sequences of these specimens were identical to the GenBank sequences of P. gregarius (Fig. 3C). In addition, individual variation in the number of externa of rhizocephalans has also been reported previously (Reinhard, 1942; Shiino, 1943; Høeg & Lützen, 1985).

In phylogenetic trees (Figs. 3A–3C), Polyascus cf. gregarious was clustered with P. planus that is commonly found in Japan and Taiwan. These two species are similar in having a flat-cordiform shaped externa, but different in some aspects of morphology and host species: the former has an elevated mantle and single externa, while the latter has an underdeveloped mantle and multiple externae (Boschma, 1933). In addition, the Varunidae crabs (H. sanguineus and H. takanoi) are used as P. cf. gregarious hosts, whereas the Grapsidae crabs (Grapsus albolineatus and Metopograpsus messor) are known as P. planus host (Tu, Chan & Jeng, 2009). Morphological and host range variation among rhizocephalan species has been reported by previous studies (Høeg & Lützen, 1985; Jung, Yoshida & Kim, 2019), and thus further studies with broader taxon sampling of P. gregarious and P. planus are needed to confirm an accurate species delimitation in their morphology and host range.

Phylogenetic relationships among rhizocephalan species

Since only Sacculinidae and Polyascidae species were found in this study, we focused on phylogenetic relationships among rhizocephalan species in these two families. Totals of 34 cox1 (555 bp), 33 16S rDNA (474 bp), and 35 18S rDNA (1002 bp) sequences were used for phylogenetic analysis, and the resulting ML and Bayesian trees were consistent with each other in that Sacculinidae and Polyascidae were monophyletic (Figs. 3A–3C). In all phylogenetic trees, the sequences of Korean rhizocephalans species nested and/or clustered with sequences of the same species retrieved from GenBank (Figs. 3A–3C).

Parasacculina sp. 1 and 2 were recognized as new species candidates because they did not show sister relationships with other Parasacculina species (Figs. 3A–3C). In the 16S and 18S DNA trees, they were placed at different positions and separated from P. leptodiae and P. sinensis, which share the host family and have similar morphological characteristics (Figs. 3B and 3C). Parasacculina sp. 1 and Parasacculina sp. 2 formed a group with P. shiinoi, but their 16S pairwise sequence divergences from P. shiinoi were substantial, ranging from 18.2–20.9% for Parasacculina sp. 1 and 14.4–14.8% for Parasacculina sp. 2. This group (Parasacculina sp. 1, Parasacculina sp. 2, and P. shiinoi) was separate from P. leptodiae in the 16S tree, whereas it was basal to the remaining Polyascidae species, including P. leptodiae and P. sinensis, in the 18S tree. In the cox1 tree, Parasacculina sp. 1 formed a well-defined sister group to Polyascus species (P. cf. gregarious and P. planus), whereas Parasacculina sp. 2 was grouped with the Korean isolate of P. shiinoi showing 18.1% sequence divergence (Fig. 3A). Interspecific sequence differences of the two new species candidates from other Polyascidae species were 18.1–32.1% for cox1, 14.0–28.8% for 16S rDNA, and 1.9−4.7% for 18S rDNA. In contrast to the high interspecific sequence divergences discovered, there were no individual variations in cox1, 16S, and 18S rDNA sequences among Parasacculina sp. 1 specimens.

The three Sacculina species (S. imberbis, S. pilosella, and S. pinnotherae) clustered with Parasacculina species (Figs. 3A–3C): S. imberbis grouped with S. pinnotherae, that is sister to other Parasacculina species based on analysis of cox1 (P. yatsui, P. granifera) and 18S rDNA (P. yatsui, P. sinensis, P. leptodiae) sequences (Figs. 3A, 3C). S. pilosella formed a sister group to P. compressa, P. oblonga, and P. yatsui in the 16S and 18S rDNA trees (Figs. 3B and 3C). Interspecific sequence differences between Sacculina species and Parasacculina species were remarkably large, ranging from 27.3–33.4% for cox1, 27.5–34.9% for 16S rDNA, and 9.2–10.1% for 18S rDNA. In contrast, intraspecific sequence divergences were very low with a maximum sequence difference of 1.0% for cox1 and 0.2% sequence difference for 16S rDNA sequences among S. confragosa individuals. All S. pinnotherae individuals had identical cox1, 16S, and 18S rDNA sequences (Figs. 3A–3C).

Discussion

In this study, we identified nine species of Korean rhizocephalans from eight host decapod species using morphological and molecular analyses. Close examination of host ranges revealed that Korean rhizocephalan species have a different host prevalence than reported for rhizocephalan species from other geographic regions. In Korea, rhizocephalans were firstly found from three decapod hosts, i.e., Hemigrapsus takanoi, Macromedaeus distinguendus, and Pugettia intermedia. We also found that most Korean rhizocephalans showed high host specificity, parasitizing only one host, except Polyascus cf. gregarius that was found on two crab species (Fig. 4). The notable differences in host range between geographic isolates (i.e., rhizocephalans from Korea and other geographic regions) might be due to geographical variation in host species diversity and abundance or insufficient information about the geographic origins of host crab species as proposed by Jung, Yoshida & Kim (2019). In addition, unlike Korean S. confragosa individuals that were all found on only one grapsid crab species, Gaetice depressus, the Japanese form is known to parasitize three crab species, G. depressus, Pachygrapsus crassipes, and Cyclograpsus intermedius. Furthermore, Japanese P. yatsui parasitizes not only G. depressus, but also P. crassipes (Tsuchida, Lützen & Nishida, 2006; Kobayashi et al., 2018), whereas the Korean form of P. yatsui was found only on Hemigrapsus sanguineus. We could not determine if other crab species including P. crassipes and C. intermedius are potential hosts of Korean S. confragosa and P. yatsui because of the limited pool of crab host species examined in this study. Extensive taxon sampling of decapod hosts and their parasitic barnacles is needed to obtain a complete understanding of the host ranges of rhizocephalan barnacles and the distribution and prevalence of host-parasite associations.

The decapod host infestation rate of Korean rhizocephalan barnacles was much lower than that reported for Japanese species. In Japan, 35 individuals representing three rhizocephalan species were found in 354 individuals of three crab species, corresponding to an infestation rate of 9.9% (Tsuchida, Lützen & Nishida, 2006). By contrast, the infestation rate of Korean rhizocephalans was substantially lower at 1.2% on average (Table 1). Species richness and extent of host usage by parasitic barnacles are tightly correlated to the availability of host species (species diversity and abundance; Kamiya et al., 2014). Differences in the extent of host usage by rhizocephalan barnacles between the two geographic regions are likely due to differences in host species diversity and abundance, as well as the sample size of examined materials (e.g., total numbers of individuals and host species). Since we examined the prevalence of rhizocephalans on all decapod hosts (a total of 3,262 host individuals inspected), our result is likely an accurate estimate of the infestation rate. On the other hand, this prevalence difference between Korea and Japan may be originated from salinity, season, host sex and size (Mouritsen et al., 2018) or biogeographical differences (Kim et al., 2020). In a previous study, the infestation rate of Korean hermit crabs by rhizocephalans was reported to be 0.9% (Jung, Yoshida & Kim, 2019), which is similar to the infestation rate observed in this study. The unexpectedly high infestation rates (>50%) of Pachygrapsus crassipes and Arcotheres sinensis are due to strong bias from the very small sample size (one to four individuals) examined. The marine ecosystems in different geographic regions display different assemblages of barnacles (Kim et al., 2020) and thus extensive sampling of parasitic barnacles from a wide range of decapod host species is necessary to better understand their prevalence, infection intensity, and host range specificity (Mouritsen et al., 2018).

In addition to their host ranges, morphological and molecular analyses in this study provided insights into the taxonomy of Korean rhizocephalan barnacle species. Phylogenetic trees recognized four monophyletic rhizocephalan families, i.e., Polyascidae, Sacculinidae, Peltogastridae, and Peltogasterellidae, consistent with previous molecular analysis (Høeg et al., 2019) and morphology-based classification. Polyascidae is characterized by multiple externa and reproduces asexually (Glenner, Lützen & Takahashi, 2003), whereas Sacculinidae is characterized by single externa and sexual reproduction. Peltogastridae and Peltogasterellidae species mainly parasitize hermit crabs, and Peltogastridae is distinguished from Peltogasterellidae by the presence of the chitinous shield on its middle part of externae (Høeg et al., 2019). Two new species candidates in the genus Parasacculina (Parasacculina sp. 1 and Parasacculina sp. 2) were recognized based on molecular phylogenetic analyses. These species were distinct from their congeneric species, P. leptodiae and P. sinensis, based on phylogenetic analyses of mtDNA (16S rDNA) and nuclear (18S rDNA) sequences (Figs. 3B–3C) even though they are morphologically indistinguishable and were found in the same host species. These two species are genetically distinct cryptic species. Furthermore, we transferred three Korean Sacculina species (i.e., Sacculina imberbis, S. pilosella, and S. pinnotherae) to the genus Parasacculina because they grouped with Parasacculina species in mtDNA cox1, 16S, and 18S rDNA phylogenetic trees (Figs. 3A–3C). This new taxonomic replacement is consistent with previous studies that transferred several Japanese and Chinese Sacculina species to Parasacculina based on molecular evidence (Tsuchida, Lützen & Nishida, 2006; Glenner et al., 2010; Høeg et al., 2019).

Comparison of the external cuticles of Korean species with previously published morphological data provided new insight into the taxonomic status of the families Sacculinidae and Polyascidae. Although Høeg et al. (2019) showed that Sacculinidae and Polyascidae are phylogenetically distinct, the original descriptions of Polyascidae (Høeg et al., 2019) did not specify morphological characters differentiating this family from Sacculinidae. For example, Høeg et al. (2019) noted that polyascids have a smooth or almost smooth external cuticle, but some polyascid species (P. pinnotherae comb. nov. and P. yatsui) in the present study had wrinkled cuticles (Figs. 2C, 2F). In addition, Høeg et al. (2019) mentioned that Polyascus species have multiple externae, but Polyascus cf. gregarius in this study had only a single externa (Fig. 2G). These results indicate that the morphological characteristics of external cuticles, previously considered to be taxonomically valid features, are highly variable and cannot be used as diagnostic characters. Future comparative analyses of morphological characters along with molecular sequences are necessary to confirm the taxonomic status of Sacculinidae and Polyascidae and the taxonomic replacement of the three Korean Sacculina species in the genus Parasacculina.

Conclusions

In conclusion, this is the first report of the taxonomy, species diversity, and host ranges of Korean parasitic rhizocephalan barnacles based on morphological and molecular analyses. We identified nine parasitic barnacle species, including two new species candidates in the genus Parasacculina, in Korea. In addition, we found higher host specificity and lower infestation rates for Korean rhizocephalan species than reported for rhizocephalan species from other geographic regions. Nevertheless, the results of this study are based on preliminary data derived from limited taxon sampling in a narrow geographic range in Korea. Additional data from extensive samplings of parasitic barnacles from a wide range of crustacean host species are necessary to better understand the taxonomy, prevalence, host usage, and phylogenetic relationships of rhizocephalan species.

Jongwoo Jung, Hee-seung Hwang (Ewha Womans University), Hyun Kyong Kim (Honam National Institute of Biological Resources), Jin-Hyeop Jeong (Soonchunhyang University), and Heesoo Kim (Korea Polar Research Institute) helped sample the materials analyzed in this study. Sang-Hui Lee helped identification of P. intermedia, the host of P. pilosella. We thank Iriomote Station, Tropical Biosphere Research Center, University of the Ryukyus and Marine and Coastal Research Center, Ochanomizu University for providing laboratory facilities, and Mr. Yuki Miyaoka and his friends for generous help in sampling.

Additional Information and Declarations

Competing Interests

Author Contributions

Data Availability

The authors declare there are no competing interests.

Jibom Jung and Joong-Ki Park conceived and designed the experiments, performed the experiments, analyzed the data, prepared figures and/or tables, authored or reviewed drafts of the paper, and approved the final draft.

Ryuta Yoshida and Damin Lee performed the experiments, authored or reviewed drafts of the paper, performed field work, and approved the final draft.

The following information was supplied regarding data availability:

Nucleotide sequences are available in GenBank (mtDNA cox1: MZ216468–MZ216513; 16S: MZ215675–MZ215720; 18S rDNA: MZ215557–MZ215602).

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
