# Peer review of "Morphological and molecular analyses of parasitic barnacles (Crustacea: Cirripedia: Rhizocephala) in Korea: preliminary data for the taxonomy and host ranges of Korean species"

_PeerJ, doi:10.7717/peerj.12281_

## Round 0.1 · original submission · Major Revisions

I agree with other reviewers that this manuscript is worthy of publication in PeerJ.

Reviewer 1 has requested that you cite specific references. You may add them you believe they are especially relevant. However, I do not expect you to include these citations, and if you do not include them, this will not influence my decision.

I have no further comments as all of my concerns have already been voiced out by the reviewers.

Reviewer 1 ·

Basic reporting

This manuscript examine the diversity and taxonomy of Rhizocephalan barnacles in Korea

Experimental design

The molecular analysis and morphological examination is fine.

Validity of the findings

This is an important paper to provide baseline Rhizocephalan barnacles in Korea and I would support accept after major revision as my comments below.

Additional comments

The MS can be accepted after major revision.

1) Introduction, first paragraph, the author describe the basic biology of Rhizocephalan. I would suggest the author also cite the lastest molecular based taxonomy of Rhizocephalans which included in Chan et al. (2021). This can be added in line 34: Latest molecular based taxonomy of Rhizocephalan barnacles was included in Chan et al. (2021).

Chan, B. K. K., N. Dreyer, A. S. Gale, H. Glenner, C. Ewers-Saucedo, M. Pérez-Losada, G. A. Kolbasov, K. A. Crandall, and J. T. Høeg. 2021. The evolutionary diversity of barnacles, with an updated classification of fossil and living forms. Zoological Journal of the Linnean Society zlaa 160. https://doi.org/10.1093/zoolinnean/zlaa160

2) Line 45, The authors reported diversity of Rhizocepahala was reported in different geographical regions. There is also diversity of rhizcocephala in Taiwan. I would suggest to cite cite Yoshida et al. 2012.

Yoshida R, Hirose M, Mok HK, Hirose E. 2012. The first records
of peltogastrid rhizocephalans (Crustacea: Cirripedia:
Rhizocephala) on hermit crabs (Paguroidea) in Taiwan and
differences in prevalences among collection sites. Zool Stud
51:1027–1039

3) Line 46, the author stated “little is known about the species diversity, distribution, taxonomy, and host range of rhizocephalan parasitic barnacles in Korea.” The author should add one more sentence to state “Jung et al. (2009) described xx species of Peltogastrid barnacles from xx hermit crabs in Korea.

4) Line 47, I would suggest to add that in Korea, there are three major oceanographic ecoregions, including the Yellow Sea, East Sea and East China Sea ecoregions, in which thoracican barnacles have different assemblages among these ecoregions (Kim et al. 2021). Then the author can follow that to survey Rhizocephalans in different ecoregions in Korea.

Kim HK, Chan BKK, Lee S-K, Kim W (2020). Biogeography of intertidal and
subtidal native and invasive barnacles in Korea in relation to oceanographic current
ecoregions and global climatic changes. Journal of the Marine Biological Association of the United Kingdom 100, 1079–1091. https://doi.org/10.1017/S0025315420001009

5) Line 54, the author should state the 16 sites in different ecoregions in Korea following Kim et al. 2021. This can be Figure 1A, then Figure 1B can be the distribution map. The sites where no rhizocephalan was found should also be indicated.

6) In discussion, the author should state their molecular tree pattern which showed three major monophyletic clades, the Polyascidae, Sacculinidae and Peltogastidiae. The author should state the Polyascidae is characterized by multiple externa and can reproductive by asexusal reproduction (Glenner et al. 2003), Sacculinidae – single externa, sexual reproduction. Peltogastridiae – mainly in hermit crabs, elaborate more on its features.

7) The authors should plot the vertical bar charts as additional figure, each bar represent one Rhizocephala species, the bar showing different percentage proportion of decapod or hermit crab species. This can provide a good ecological figure on host usage of different species.

8) Since Jung et al 2019 has conducted the survey of Peltogastid barnacles in Korea, this pattern should be also added together in the map in Figure 1 with the present data (but use different colour for the data in Jung et al 2019), to give an overall distribution pattern of Rhizocephalan barnacles in Korea.

·

Basic reporting

This is an exemplary paper on how rhizocephalan taxonomy and fainistics should be done. The team, including the experienced second author have done a fine job. I have only a very few comments and corrections
1) The English is clear and I have only corrected inb one or two places
2) The Literature is well treated and referred to. Nevertheless, I miss reference in appropriate places to two important articles:
Kobayashi M; Wong YH; Oguro-Okano M; Dreyer N; Høeg JT; Yoshida R; Okano K.. 2018. Identification, characterization, and larval biology of a rhizocephalan barnacle, Sacculina yatsui Boschma, 1936, from northwestern Japan (Cirripedia: Sacculinidae). J Crustacean Biol 38(3): 329–340. doi:10.1093/jcbiol/ruy020
Øksnebjerg, B. 2000. The Rhizocephala (Crustacea: Cirripedia) of the Mediterranean and Black Seas: taxonomy, biogeography, and ecology. Israel Journal of Zoology, 46: 1–102.
Especially Oksnebjerg seems to stand as the "above all" general treatment on nmodern rhizocephalan taxonomy, al be it only on European species.

Also: in the article below the authors will find the largest ever survey of a rhizocephaælan on its host and in a classic system. It does not directly inform the present MS but puts the data in a perspective and supports what the authors state: That large samples from large areas and over time is needed to fully understand host specificity and prevalences....
Mouritsen K; Geyti S; Lützen J; Høeg JT; Glenner H. 2018. Population dynamics and development of the rhizocephalan, Sacculina carcini, parasitic on the shore crab Carcinus maenas. Diseases of Marine Organisms 131: 199-211. https://doi.org /10.3354/dao03290



The article structure is fine. I am very happy with the material presented in both morphological figures, distribution maps ands the several informative tables. This is how things should be done on these parasites

The article is indeed self-contaioned. This is MORE than a simple taxonomy paper, since its summarizes the fauna of an entire region and raises ideas and questions for future research.

Experimental design

Nothing to argue with

Validity of the findings

I agree with all conclusions, including the taxonomic steps. It ois very satisfactry that the authors use a phylogenetic approach to taxonomy and thus assist in resolving the issue of Sacculia and Parasacculina. I think Hoeg et al. (2019) predicted that many species of Sacculina might be in the future moved to Polyascidae-Parasacculina and now we see the fiurst such example. well done!

Additional comments

1) I wonder why the authors did not take the full step and described the two new species of Parasacculina as such? I accepot their hesitancy and that it may be either "in the offing" or that they wait until they have further data (could be hostology as I know is exercized by author YOSHIDA). So please STATE why you do not formally describe those species

2) Multiple (colonial) externae. As the authors know, there are two types of multiple externae. Those that are internally interconnected and belong to one individual (Colonialism) and thus derive form a single infection. And those that represent individual infection events and thereofre separate individuals. Normally COLONIAL species are always with multiple externae. BUT here they find that Polyascus greegarius can be single on the host here studied. This raises new perspectives about this biology. Could it be that colonialism is a plastic character that depends in part on host species? I thereofre encourage the authors to briefly mention this even if they cannot offer solutions!

Specific comments

line 31: Remember rhizocephalans also occur on other crustacea, although this concerns only the "former akentrogonids"

Line 42: "In Europe" I do not understand. I guess this should be deleted or specified as "Far East"?
line 97: => externae
line 203: mantle? I think they mean "mantle aperture" or the mantle part that carries the "aperture". Please be clear on this

Line 257: => sinensis

Linbe 301-302: I am no sure here. Do you mean you are FIRST to describe rhizocephalans form these two crabs?

line 312: Here is a good place to also cite Kobayashi et al. (see above)

Lines 359-361: This may be a place to briefly discuss the issue of multiple externae and its intraspecific variation (or do it in detail before this)

Reviewer 3 ·

Basic reporting

This manuscript presented good and significant data for the study concerning this Sacculinidae parasite infection. The experiments were well-planned and the finding of this study is new. Interestingly, there are many sacculinidae infections from various species found in the sampling sites. By using proper taxonomical analysis based on morphology and combining with three molecular markers, the coding region from the nuclear and mitochondrial genome, and the non-coding region from the mitochondrial genome to construct the phylogenetic tree is one of the best ways to group the organism properly to the right taxa. The selected genes were from uniparental lineage, giving more genetic information for comparison.

Experimental design

The experiment was properly designed with complete morphological and molecular analysis to characterize the taxa of Sacculinidae. These approaches support the main objectives of the study very well. However, using MEGA 10 software to construct the maximum-likelihood phylogenetic tree is unpopular but it is still acceptable. I have no problem with this matter. However, most of the researchers prefer to use RaxML software to construct the phylogenetic tree.

Validity of the findings

The findings are reliable and novel based on the approaches used to characterize the sacculinidae species and group them into the right taxa. These promising results might be useful in future studies of Sacculinidae parasites. The conclusions are well described in this manuscript.

Additional comments

The manuscript is well written and presented. This research is really comprehensive and novel. However, there are some minor grammatical mistakes in the manuscript. Please correct all the grammatical mistakes in the manuscript. The author should also improve the map of the sampling sites by providing the name of the sea around South Korea.

---

## Round 0.2 · Minor Revisions

Please kindly address the minor comments raised by the reviewer.

Reviewer 1 ·

Basic reporting

see below

Experimental design

see below

Validity of the findings

see below

Additional comments

I think my previous comments were fully addressed. There are two minor points:

1) Polyascus - it is identified as P. gregarious in Korea. There is another species Polyascus plana commonly in Japan including Okinawa and Taiwan. From the phylogenetic tree in this MS, although P. plana and P. gregarious are separated in two clades, but the two clades are very close each other. I wonder it may be population level differences in P. plana. Any comments in this aspect? I think the author should add in one paragraph to address this, if it is P. gregarious, there should be morphological differences from P. plana which can add into the remarks.

2) Jung et al. in the reference list - it should be Zoological Studies 58: 33 - the article number is missing in the MS.

---

## Round 0.3 · accepted · Accept

The authors addressed all concerns accordingly.

Reviewer 1 ·

Basic reporting

see below

Experimental design

see below

Validity of the findings

see below

Additional comments

The comments were addressed and it can be accepted for publication.